# Mechanisms Underlying the Inhibition of Tyrosine Kinase Inhibitor-Induced Anorexia and Fatigue by Royal Jelly in Renal Cell Carcinoma Patients and the Correlation between Macrophage Colony Stimulating Factor and Inflammatory Mediators

**DOI:** 10.3390/medsci8040043

**Published:** 2020-10-09

**Authors:** Tsutomu Yuno, Yasuyoshi Miyata, Yuta Mukae, Asato Otsubo, Kensuke Mitsunari, Tomohiro Matsuo, Kojiro Ohba, Hideki Sakai

**Affiliations:** Department of Urology, Nagasaki University Graduate School of Biomedical Sciences, Nagasaki 852-8501, Japan; tsutomu.y.october.8@gmail.com (T.Y.); ytmk_n2@yahoo.co.jp (Y.M.); a.06131dpsc@gmail.com (A.O.); ken.mitsunari@gmail.com (K.M.); tomozo1228@hotmail.com (T.M.); ohba-k@nagasaki-u.ac.jp (K.O.); hsakai@nagasaki-u.ac.jp (H.S.)

**Keywords:** royal jelly, tyrosine kinase inhibitor, renal cell carcinoma, cytokines, MSCF

## Abstract

Inflammation is a common adverse event of anti-cancer therapy. Royal jelly (RJ) modulates inflammation by regulating the levels of tumor necrosis factor (TNF)-α, transforming growth factor (TGF)-β, and interleukin (IL)-6 produced by macrophages. Macrophage colony stimulating factor (M-CSF) is a crucial regulator of macrophage activities, and we hypothesized that RJ alters M-CSF levels. In this randomized controlled trial, we investigated the association between M-CSF and adverse events in renal cell carcinoma patients treated with tyrosine kinase inhibitors (TKIs) after an oral intake of RJ (*n* = 16) or placebo (*n* = 17). The serum levels of M-CSF, TNF-α, TGF-β, and IL-6 were measured by an enzyme-linked immunosorbent assay, and their temporal changes and correlation between such changes were analyzed. The post-/pretreatment ratio of M-CSF levels was associated with anorexia after 2 weeks and fatigue after 2, 4, and 12 weeks. The M-CSF level in the RJ group was higher than that in the placebo group at the same timepoints. The TNF-α level in the RJ group was lower than that in the placebo group between 6 and 12 weeks, and the TGF-β level in the RJ group was higher than that in the placebo group; however, contrasting findings were detected after 12 weeks. Additionally, the M-CSF level was significantly correlated with the TGF-β level after 4 weeks and IL-6 level after 8 and 10 weeks. Among TNF-α, TGF-β, and IL-6, the post-/pretreatment ratio of TGF-β after 12 weeks was associated with TKI-induced anorexia, and the ratios after 10 and 12 weeks were associated with fatigue. Our results demonstrated that an oral intake of RJ suppressed anorexia and fatigue via complex mechanisms associated with inflammation-related factors, such as M-CSF and TGF-β in renal cell carcinoma patients treated with TKIs. In addition, we newly found that such RJ-related effects were dependent on the treatment duration.

## 1. Introduction

Chronic inflammation is closely associated with the pathogenesis of various types of chronic diseases, and many investigators have paid special attention to the preventive effects of a variety of nutraceuticals that regulate disease-related inflammation [1,2]. Additionally, numerous investigators have suggested that complex immune responses, regulated by various types of immune cells, cytokines, and pro-/anti-inflammatory mediators, play a crucial role in cancer-related inflammation [3,4]. Interleukin (IL)-6, tumor necrosis factor (TNF)-α, and transforming growth factor (TGF)-β are also closely associated with several cancer-related symptoms and adverse events of anti-cancer therapy, such as cachexia and fatigue [5,6,7]. Thus, understanding the regulatory mechanisms underlying chronic inflammation is important for the management of anti-cancer therapy-associated unpleasant symptoms experienced by cancer patients.

Renal cell carcinoma (RCC) is one of the major types of urological cancer, and metastatic RCC is recognized as a fatal disease. At present, various types of anti-cancer agents are commonly used for patients with advanced/metastatic RCC, and tyrosine kinase inhibitors (TKIs) are the most prevalent first-line therapy [8]. Conversely, one of the most important disadvantages of TKIs is the relatively high frequency of adverse events, which are occasionally severe. Therefore, information on the causative and pathological mechanisms underlying such unpleasant TKI-induced symptoms is essential before treatment strategies can be discussed. Similar to those in most cancers, the activities and accumulation of various immune cells, for example, lymphocytes, natural killer cells, and neutrophils, within the tumor microenvironment are considered important determinants of cancer-related inflammation in RCC [9,10]. In addition, macrophages within the tumor microenvironment (i.e., tumor-associated macrophages or TAMs) play important roles in cancer-related inflammation in RCC patients [9,11,12,13]. A variety of cytokines, including IL-6, TNF-α, and TGF-β, are produced from TAMs in RCC [11,14]. As mentioned above, these cancer-related inflammatory mediators are closely associated with cancer-related complications [5,6,7]. Therefore, the activities of macrophages are speculated to be correlated with a variety of cancer-related symptoms in RCC patients. Macrophage colony stimulating factor (M-CSF) is a key modulator of the activation, migration, and survival of monocytes/macrophages under various pathological conditions [15,16]. Therefore, M-CSF is speculated to affect the production of IL-6, TNF-α, and TGF-β via the activation of macrophages, and thereby associated with cancer-related inflammation symptoms. However, little information exists regarding the correlation between M-CSF and IL-6, TNF-α, and TGF-β in humans.

Royal jelly (RJ) is secreted by the hypopharyngeal and mandibular glands of worker honeybees and is considered safe for human consumption and good for overall health. The biological effects of RJ on macrophage activity are not fully understood; however, several components of RJ, such as 10-hydroxy-trans-2-decanoic acid (10H2DA) and adenosine N1-oxide, have been reported to affect the activation and function of macrophages both in vivo and in vitro [17,18]. Additionally, M-CSF is a key modulator of the biological and pathological activities of macrophages [15,16]. Therefore, we hypothesized that RJ administration modulates macrophage activities by regulating M-CSF levels. Accordingly, the relationship between the clinical usefulness of RJ intake (i.e., suppression of anorexia and fatigue) and change in M-CSF levels was analyzed.

We previously reported that an oral intake of RJ suppresses anorexia and fatigue in RCC patients treated with TKIs [19]. However, the detailed mechanisms of such RJ-related functions were unclear in said study. We also found that the post-/pretreatment ratio of TGF-β decreased after an oral intake of RJ, and this change was associated with the incidence of anorexia and fatigue in a similar study population [7]. In addition, this previous study revealed that the oral intake of RJ decreased the post-/pretreatment ratio of TNF-α but not that of IL-6 and these changes were not associated with anorexia and fatigue. However, these results were obtained by measuring serum levels at baseline and 3 months after the commencement of treatment. In short, no data were obtained regarding the temporal change of IL-6, TNF-α, and TGF-β after oral RJ intake. In addition, the influence of macrophages on such RJ-induced functions was not investigated. Therefore, in the current study, we investigated the temporal changes in serum M-CSF levels after an oral intake of RJ in RCC patients treated with TKIs. Moreover, the correlation between the change in post-/pretreatment ratio of serum M-CSF and the ratios of inflammation-related cytokines (i.e., IL-6, TNF-α, and TGF-β) was investigated. Taken together, this current study presented new information regarding the complex mechanisms of RJ-related prevention of anorexia and fatigue; for example, the clinical and biological roles of M-CSF and temporal changes in inflammation-related factors in RCC patients treated with TKIs were demonstrated.

## 2. Materials and Methods

### 2.1. Ethics

The study protocol was approved by the Human Ethics Review Committee of Nagasaki University Hospital (Nagasaki, Japan; No. 15102604-2; registered as UMIN000020152). In the original protocol, we tried to recruit 60 patients in this clinical study. However, unfortunately, it failed to reach the goal. All experiments complied with the principles embodied within the Declaration of Helsinki and patients provided written informed consent to participate in all aspects of the study.

### 2.2. Patients and Study Protocol

The study population and protocol were similar to those of our previous reports [7,19]. Briefly, 33 patients with metastatic RCC who had been assigned to commence first-line TKI therapy were enrolled in this study. Of the study population, 27 patients (87.9%) were judged to have clear cell RCC; however, there was no significant difference in the ratio of clear cell RCC between the placebo group and the RJ group (94.1% and 81.3%, respectively; *p* = 0.446) [19]. The placebo (starch) and RJ, which were similar in taste, smell, size, shape, and color, were provided by the Yamada Agriculture Center Inc. (Okayama, Japan). In our protocol, 2400 mg/day of RJ (capsules containing 800 mg RJ were administered orally three times per day) was administered orally for three months. This study was a randomized, double-blinded, placebo-controlled clinical trial. General conditions and clinical symptoms, including anorexia and fatigue, were examined, and blood samples were collected every two weeks. Evaluations of anorexia and fatigue were performed according to the National Cancer Institute Common Terminology Criteria for Adverse Events, version 5.0 as well as previous reports [7,19]. In short, when any symptom was found, even if it was a grade 1 symptom, the patient was judged as having the “presence” of anorexia or fatigue. Blood samples were subjected to laboratory analyses routinely performed in patients receiving TKI therapy. In addition, the serum levels of M-CSF, TNF-α, TGF-β, and IL-6 were quantified by enzyme-linked immunosorbent assay (R&D systems, Inc., Minneapolis, MN, USA) before and after 2, 4, 6, 8, 10, and 12 weeks of treatment. Regarding the TKIs administered, the starting doses of sunitinib, pazopanib, axitinib, and sorafenib were 50, 800, 10, and 800 mg/day, respectively. Information regarding the patient selection process, experimental agents, and treatments is detailed in previous reports [7,19]. The pretreatment data, collected in the morning of the first treatment day, were evaluated before the oral intake of RJ and TKIs.

### 2.3. Statistical Analyses

Results are expressed as the median and interquartile range. The Mann–Whitney U test was used to compare the continuous variables, and Pearson’s correlation coefficient (r) was used to evaluate the relationship between these variables. The coefficient of variation was calculated as a ratio of the standard deviation to the mean. All the statistical analyses were two-sided and were performed on a personal computer using the statistical package StatView for Windows (version 5.0, Abacus Concept, Inc., Berkeley, CA, USA). Significance was set at *p* < 0.05.

## 3. Results

### 3.1. Patient Demographics

Our study population consisted of 23 males and 10 females, and the median age at the time of treatment was 68 (range: 54–79) years. With regard to pathological features, 81.8% (27/33 patients) and 72.7% (24/33 patients) were diagnosed with high-grade disease (Fuhrman grade 3 and 4) and high pT stage (pT3 and 4), respectively. Finally, no significant differences were observed between the placebo (*n* = 17) and RJ group (*n* = 16) regarding age (*p* = 0.101), sex (*p* = 0.909), Eastern Cooperative Oncology Group performance status (*p* = 0.598), grade (*p* = 0.425), pT stage (*p* = 0.201), lymph node metastases (*p* = 0.881), and distant organ metastases (*p* = 0.325). Additionally, no significant difference was observed between the placebo and RJ group regarding the type of TKI administered (*p* = 0.539) [7,19].

### 3.2. Changes in the Post-/Pretreatment Ratio of M-CSF after RJ Intake and Its Clinical Roles

As presented in Figure 1A, the post-/pretreatment ratio of serum M-CSF levels in the RJ group was significantly higher than that in the placebo group at 2 and 4 weeks after commencing treatment (*p* = 0.014 and 0.007, respectively). Conversely, no significant difference was detected in the ratio of M-CSF between 6 and 10 weeks; however, a remarkable difference was observed at 12 weeks (*p* < 0.001; Figure 1A).

Similarly, the ratios at 2, 4, and 12 weeks in patients without fatigue were higher than those experiencing fatigue (*p* = 0.006, 0.027, and 0.017, respectively; Table 1). With regards to anorexia, however, a significant difference was found in the ratio of M-CSF at 2 weeks (*p* = 0.004), but not for the other timepoints, including 4 and 12 weeks.

### 3.3. Changes in the Post-/Pretreatment Ratio of Inflammation-Related Markers after RJ Intake

Although the post-/pretreatment ratio of serum TNF-α levels between the placebo and RJ group demonstrated no significant difference at 2 and 4 weeks, the ratio in the RJ group was significantly lower than that in the placebo group at 6, 8, and 10 weeks (*p* = 0.048, 0.015, and 0.001, respectively; Figure 1B). On the other hand, our previous report showed that the ratio at 12 weeks showed a significant difference between these two groups (*p* = 0.007) [7]. Additionally, the post-/pretreatment ratio of serum TGF-β levels in the RJ group was significantly higher than that in the placebo group at 4 weeks (*p* = 0.014; Figure 1C). However, the opposite trend was observed at 12 weeks in our previous report; briefly, the post-/pretreatment ratio of serum TGF-β levels in the RJ group was significantly lower (*p* = 0.013) than that in the placebo group at 12 weeks [7]. In contrast, no significant difference was observed in the ratio of serum IL-6 levels between the placebo and RJ groups for all treatment periods (Figure 1D) including at 12 weeks [7].

### 3.4. Correlation between the Ratios of Inflammation-Related Markers and Anorexia or Fatigue

Correlation analysis of the post-/pretreatment ratios of serum TNF-α, TGF-β, and IL-6 levels with fatigue or anorexia over 2–10 weeks (Table 2) revealed that the post-/pretreatment ratio of serum TGF-β levels at 10 weeks in patients with fatigue (median = 0.94; interquartile range; IQR = 0.78–1.04) was significantly higher (*p* = 0.020) than that in patients without fatigue (median = 0.68; IQR = 0.55–0.81). In addition, with regards to fatigue, the ratio of TGF-β at 8 weeks demonstrated a similar trend; however, this difference was not significant (*p* = 0.083; Table 2). In contrast, IL-6 and TNF-α levels were not associated with anorexia and fatigue at any timepoint (Table 2).

### 3.5. Correlation between the Ratio of M-CSF and IL-6, TNF-α, or TGF-β Ratios

The post-/pretreatment ratio of serum M-CSF levels was significantly correlated with that of TGF-β levels at 4 weeks (r = 4.03, *p* = 0.020) but not for the other timepoints, including 12 weeks (Table 3). Similarly, there was no significant correlation between the post-/pretreatment ratio of serum M-CSF levels and that of TNF-α levels for any of the treatment periods, including 12 weeks (Table 3). In contrast, the ratios of M-CSF levels were closely associated with the ratios of IL-6 levels at 8 and 10 weeks (r = 0.54, *p* = 0.001 and r = 0.62, *p* < 0.001, respectively; Table 3).

## 4. Discussion

The present study showed that an oral intake of RJ influences serum M-CSF levels in RCC patients treated with TKIs. To our knowledge, this is the first study to demonstrate the relationship between RJ and M-CSF under physiological and pathological conditions. We also found that the oral intake of RJ significantly increased serum M-CSF levels after short-term treatment (2 and 4 weeks). Another study showed that two weeks of RJ administration led to changes in the serum levels of IL-1β, IL-10, and TNF-α in a rat model of colitis [20]. Moreover, the serum levels of angiotensin II, endothelin-1, and TGF-β were influenced after the administration of bee products, including RJ, for 5 weeks in spontaneous hypertensive rats [21]. Thus, one may assume that short-term RJ treatment can affect the serum levels of various inflammation-related modulators in vivo. However, while the difference in serum M-CSF levels between the placebo and RJ groups was not statistically significant after 6 weeks of treatment, the difference detected was maximal after 12 weeks of treatment. We could not justify this finding; therefore, we speculate that RJ regulates M-CSF production via complex mechanisms.

Our results showed that the oral intake of RJ affected serum TNF-α levels. Unfortunately, there is no report regarding the influence of RJ on the serum level of TNF-α in human subjects. However, in a rat colitis model, the serum level of TNF-α was significantly lower in the RJ treatment group (*p* < 0.05) than in the control [20]. Similarly, other investigators demonstrated that serum TNF-α levels were significantly lower in rats treated with cyclophosphamide and RJ than in those treated with only cyclophosphamide [22]. In addition, previous animal studies have indicated that RJ decreases the tissue levels of TNF-α in various pathological conditions, including cadmium chloride-induced testicular dysfunction, bleomycin-induced pulmonary fibrosis, and ethylene glycol-induced renal inflammation [23,24,25]. These reports support our result that the oral intake of RJ decreases serum TNF-α levels. On the other hand, we also found that the post-/pretreatment ratio of serum TNF-α levels in the RJ group was significantly lower than that in the placebo group after 6 weeks of treatment. These results suggest that RJ may suppress TNF-α production in a time-dependent manner and that its activity reached a significant level after a short period (after 2 and 4 weeks) of treatment.

Our findings on changes in serum TGF-β levels by the oral intake of RJ were interesting. Briefly, the post-/pretreatment ratio of TGF-β level in the RJ group was significantly lower than that in the placebo group after 12 weeks of treatment; however, the opposite result was found at 4 weeks. While no studies exist regarding the change in TGF-β levels after RJ administration in cancer patients, in one study, RJ increased TGF-β production in ultraviolet B (UVB)-irradiated human skin fibroblasts [26]. In addition, 10H2DA, which is a major extract of RJ, was reported to stimulate TGF-β production in normal human dermal fibroblasts [27]. In contrast, although TGF-β levels increased in the bronchoalveolar lavage fluid in a rat model of bleomycin-induced pulmonary fibrosis, RJ administration reversed such pathological alterations [24]. Additionally, the author emphasized that RJ-related activities were dose-dependent and were observed after long-term treatment with RJ [24]. Other investigators demonstrated that the honeybee product, Bao-Yuan-Ling, composed of propolis, RJ, and bee venom, decreased serum TGF-β level in spontaneous hypertensive rats [21]. Conversely, one study suggested that honey and major RJ protein 1 had no significant effect on TGF-β production in normal human dermal fibroblasts [28]. Therefore, TGF-β production induced by RJ was speculated to be dependent on the types of cells/tissues and the extracts contained in the RJ. In addition, we assume that the RJ treatment period and related circumstances are important determinants of TGF-β production because previous studies demonstrated that the RJ level was increased in vitro and decreased in vivo [21,26,27,28]. Thus, our observation that the oral intake of RJ increases serum TGF-β levels after short periods (2–4 weeks) but decreases it after long periods (10–12 weeks) of treatment seems to concur.

It is not yet clear whether there is a direct correlation between M-CSF and TNF-α, TGF-β, or IL-6. Histologically, activated macrophages are divided into two phenotypes, namely pro-inflammatory (M1) and anti-inflammatory (M2) macrophages, and it has been established that M-CSF induces macrophage polarization toward the anti-inflammatory (M2) phenotype. M2 macrophages secrete anti-inflammatory cytokines, including TGF-β [15]. In line with these reports, we found that changes in serum M-CSF levels were positively correlated with those of serum TGF-β levels after 4 weeks of treatment. We believe that M2-like macrophages, activated via an increased production of M-CSF, secrete higher amounts of TGF-β when RJ is administered for short durations (2–4 weeks). In contrast, no significant correlation was noted between M-CSF and TGF-β after 12 weeks of treatment, even though the post-/pretreatment ratio of serum TGF-β level was significantly lower in the RJ group than in the placebo group at the same timepoint. We speculate that RJ modulates TGF-β production by mechanisms that are independent of M-CSF after long-term treatment (12 weeks). Figure 2 represents the pathological roles of inflammation-related mediators and the role of oral RJ administration in TKI-induced anorexia or fatigue.

In the present study, we found that an oral intake of RJ decreased post-/pretreatment ratio of serum TNF-α levels after 6 weeks of treatment. However, a direct significant correlation was not found between TNF-α and M-CSF levels for all treatment periods. TNF-α is secreted from M1 macrophages polarized by granulocyte-M-CSF [15]. This finding supports our notion that RJ suppresses TNF-α production by an M-CSF-independent mechanism. In contrast, we also found that changes in serum M-CSF levels are strongly associated with changes in serum IL-6 levels. Similar to TNF-α, IL-6 is reported to be secreted from M1 macrophages [15]. This does not seem to match our results regarding the relationship between M-CSF and IL-6 levels. A variety of M2 macrophage subtypes (e.g., M2b) can produce high levels of IL-6 [15,29]. Although studies on M2b macrophages and M-CSF do not yet exist, there is a possibility that M-CSF correlates with IL-6 secretion via the polarization of M2b macrophages. M2a macrophages, which cannot produce IL-6 when treated with an immunocomplex containing IgG4, promote a change toward the M2b phenotype [30]. Moreover, the addition of M-CSF + GM-CSF to M2 macrophages previously polarized with M-CSF leads to the stimulation of inflammatory cytokines [31]. Thus, the secretory function of M2 macrophages is dependent on co-existing factors and circumstances. Unfortunately, we are not certain how RJ affects this complex network of interactions underlying macrophage functions. 

This study has several major limitations. Firstly, the number of patients included was relatively small. However, we think that bias was minimal because we employed a double-blind, randomized study design. Secondly, there were differences in the patients’ backgrounds. There is a possibility that the pharmacological effects of TKIs could influence the changes in inflammation-related factors, although we confirmed that there is no significant correlation between the post-/pretreatment ratio of the serum levels of M-CSF, TNF-α, TGF-β, or IL-6 and relative dose intensity in our study population. TKIs are still one of the major treatments for RCC patients, and many investigators pay special attention to cost, safety, and new treatment strategies using TKIs [32,33,34]. In addition, recent clinical trials showed that RJ improve various pathological symptoms [35,36]. Therefore, we believe that our results are useful to discuss such issues in RCC patients treated with TKIs. When discussing the pathological and biological roles of macrophages, we should note the objective opinion against the classification of M1 and M2 macrophages [29]. In addition, TAM turnover, function, and ontogeny are regulated by complex mechanisms and are not fully understood [37]. Therefore, we need to pay attention to the fact that the pathological roles of M-CSF in macrophages are not clear in cancer tissues.

## 5. Conclusions

An oral intake of RJ modulates serum M-CSF levels in RCC patients treated with TKIs, and these changes in M-CSF are associated with TKI-induced anorexia and fatigue. Temporal changes in M-CSF, TNF-α, TGF-β, and IL-6 levels after RJ intake are dependent on the treatment duration. RJ inhibits TKI-induced anorexia and fatigue via complex mechanisms associated with M-CSF and TGF-β, which also vary with the treatment duration. Although changes in serum M-CSF levels caused by RJ positively correlated with those of IL-6, the clinical significance of this correlation is not clear. Further long-term studies with larger study populations are required to clarify the relationship between RJ and chronic inflammation-related symptoms in RCC patients.

## Figures and Tables

**Figure 1 medsci-08-00043-f001:**
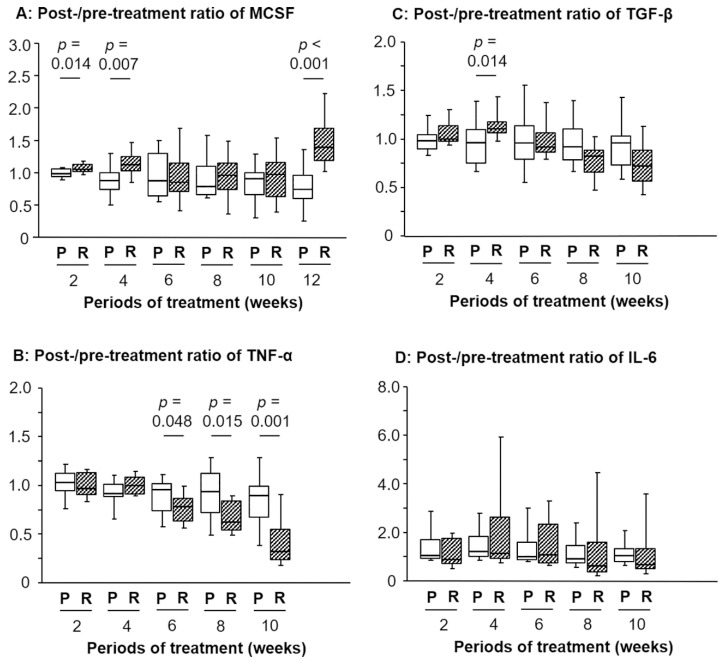
Temporal changes in the post-/pretreatment ratio of the serum levels of macrophage colony stimulating factor (M-CSF), tumor necrosis factor (TNF)-α, transforming growth factor (TGF)-β, or interleukin (IL)-6 after oral intake of royal jelly (RJ). A significant difference was detected in the ratios between the placebo (P) and R group regarding: M-CSF after 2, 4, and 12 weeks (**A**), TNF-α from 6 to 10 weeks (**B**), and TGF-β after 2 weeks (**C**). No significant difference was observed in the IL-6 ratios between the P and R group (**D**). The 12-week data of TNF-α, TGF-β, and IL-6 are shown in our previous report [7].

**Figure 2 medsci-08-00043-f002:**
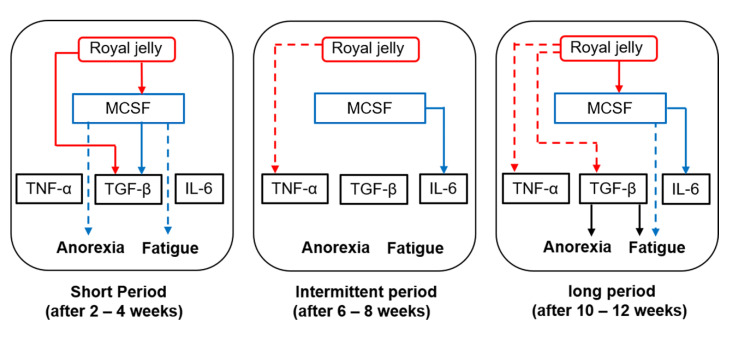
Interactions between royal jelly (RJ), macrophage colony stimulating factor (M-CSF), and inflammatory mediators after short-, intermediate-, and long-term treatment with RJ. Red lines—effects of RJ; blue lines—effects of M-CSF; black lines—effects of transforming growth factor (TGF-β). Solid lines and dotted lines refer to an increase and decrease in post-/pretreatment serum levels, respectively.

**Table 1 medsci-08-00043-t001:** Relationship between post-/pretreatment ratio of M-CSF and anorexia or fatigue.

Weeks (W)	Anorexia		*p* Value	Fatigue		*p* Value
	Absence	Presence		Absence	Presence	
2 W	**1.07/1.03–1.12**	**1.00/0.94–1.04**	**0.004**	**1.05/1.02–1.13**	**0.97/0.93–1.06**	**0.006**
4 W	1.14/0.85–1.25	0.96/0.84–1.08	0.137	**1.14/0.77–1.25**	**0.92/0.77–1.09**	**0.027**
6 W	0.83/0.68–1.10	0.87/0.65–1.26	0.772	0.83/0.74–1.14	0.88/0.65–1.21	0.941
8 W	0.89/0.69–1.22	0.92/0.66–1.13	0.971	0.95/0.68–1.14	0.84/0.68–1.17	0.632
10 W	0.93/0.47–1.09	0.93/0.68–1.01	0.885	0.98/0.58–1.09	0.92/0.71–1.02	0.883
12 W	1.26/0.95–1.51	0.86/0.71–1.37	0.219	**1.37/1/20–1.57**	**0.86/0.66–1.10**	**0.017**

Bold values mean significant difference (*p* < 0.050). Data are presented as median/interquartile range.

**Table 2 medsci-08-00043-t002:** Post-/pretreatment ratio of inflammation-related markers and anorexia or fatigue.

Variables	Anorexia		*p* Value	Fatigue		*p* Value
	Absence	Presence		Absence	Presence	
TNF-α						
2W	1.03/0.91–1.16	0.99/0.90–1.10	0.170	1.03/0.92–1.24	1.00/0.91–1.10	0.740
4W	0.95/0.92–1.07	0.95/0.89–1.05	0.104	0.95/0.92–1.07	0.97/0.90–1.08	0.713
6W	0.79/0.61–0.99	0.89/0.76–1.00	0.470	0.79/0.64–0.98	0.89/0.72–1.01	0.320
8W	0.67/0.55–0.96	0.85/0.58–0.95	0.294	0.78/0.54–0.90	0.80/0.61–1.00	0.302
10W	0.33/0.27–0.85	0.81/0.46–0.95	0.104	0.33/0.27–0.85	0.81/0.46–0.95	0.105
12W	0.53/0.10–0.97	0.79/0.24–0.87	0.805	0.27/0.04–0.90	0.86/0.46–0.91	0.294
TGF-β						
2W	1.02/0.98–1.05	0.98/0.91–1.06	0.448	1.03/0.98–1.14	0.98/0.92–1.04	0.151
4W	1.07/0.99–1.19	1.01/0.89–1.12	0.347	1.11/1.04–1.16	1.01/0.82–1.10	0.097
6W	0.91/0.80–0.98	0.99/0.91–1.14	0.159	0.91/0.81–1.00	0.97/0.83–1.14	0.320
8W	0.87/0.65–0.92	0.88/0.75–1.03	0.588	0.82/0.63–0.90	0.89/0.80–1.10	0.083
10W	0.74/0.56–0.96	0.88/0.68–1.04	0.233	**0.68/0.55–0.81**	**0.94/0.78–1.04**	**0.020**
12W	**0.61/0.46–0.76**	**0.90/0.77–0.96**	**0.030**	**0.60/0.40–0.83**	**0.88/0.63–0.99**	**0.030**
IL-6						
2W	0.91/0.72–1.91	1.06/0.91–1.36	0.588	1.13/0.80–1.82	0.98/0.83–1.54	0.768
4W	1.04/0.90–2.44	1.32/1.08–1.91	0.347	1.89/1.05–2.62	1.18/0.88–1.85	0.286
6W	1.06/0.85–2.74	1.00/0.86–1.30	0.664	1.11/0.99–1.67	0.95/0.82–1.95	0.238
8W	0.70/0.54–1.79	0.95/0.71–1.30	0.857	0.71/0.49–1.17	0.90/0.69–1.65	0.397
10W	0.69/0.57–1.06	0.98/0.75–1.87	0.148	0.81/0.56–1.38	0.89/0.63–1.35	0.632
12W	0.81/0.53–1.12	1.01/0.44–1.63	0.426	0.73/0.44–1.18	1.01/0.50–1.63	0.854

Bold values mean significant difference (*p* < 0.050). Data are presented as median/interquartile range. Data at 12 weeks were cited from our previous report [7]. TNF; tumor necrosis factor, TGF; transforming growth factor (TGF)-β, IL; interleukin.

**Table 3 medsci-08-00043-t003:** Correlation of post-/pretreatment ratio of serum M-CSF levels with the ratios of other factors (r/*p* value).

Variables	2 Weeks	4 Weeks	6 Weeks	8 Weeks	10 Weeks	12 Weeks
TNF-α	0.19/0.301	0.13/0.461	0.13/0.475	0.05/0.774	0.07/0.685	0.11/0.532
TGF-β	0.04/0.813	**4.03/0.020**	0.15/0.385	0.11/0.553	0.22/0.219	0.28/0.115
IL-6	0.23/0.204	0.16/0.366	0.13/0.461	**0.54/0.001**	**0.62/<0.001**	0.33/0.061

Bold values mean significant difference (*p* < 0.050). TNF; tumor necrosis factor, TGF; transforming growth factor (TGF)-β, IL; interleukin.

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
