# Peer review of "Mechanisms Underlying the Inhibition of Tyrosine Kinase Inhibitor-Induced Anorexia and Fatigue by Royal Jelly in Renal Cell Carcinoma Patients and the Correlation between Macrophage Colony Stimulating Factor and Inflammatory Mediators"

_medsci, 2020, doi:10.3390/medsci8040043_

Round 1

Reviewer 1 Report

This study met most standards and should be published with minor revisions.  The main criticism is that the methods study protocol section should be improved by adding descriptions for how anorexia and fatigue were measured in patients.  Was this done by survey?

In addition, there a few issues with grammar or style:

Line 19: "we" is in a bigger font.

Table 1: The caption "W; weeks" appears on the next page (line 140).  It should be moved to be part of Table 1.  Also, a description of what the bold values mean (statistical signifcance) should be added to Table 1 and 2 and 3.

Line 175 Begin sentence with "Another study showed..." or "A previous study showed..."

Line 184 "subjects."

Line 206 "that a honeybee product,..."

Author Response

August 28th, 2020

Reviewer 1

Comments to reviewer #1:

We thank the reviewer for evaluating our manuscript. We agree with your opinions, and your advice have helped us to greatly improve the manuscript. Our responses to your comments are provided below (page and line numbers in the revised version of the manuscript are indicated). In addition, the manuscript has been proofread by a native English-speaking editor from a professional English language editing service.

Point-by-point response

  1. How anorexia and fatigue were measured in patients?

Answer:

   In this study, we evaluated the anorexia and fatigue according to the National Cancer Institute Common Terminology Criteria for Adverse Events, version 5.0. In short, when any symptom was found, even if it is a grade 1 symptom, the patient was judged as “presence” of anorexia or fatigue. We have added this information to 2.2. Patients and study protocol section (lines 11-14).

About issues with grammar or style

  1. Line19: “we” is in a bigger font.

Answer:

   We are sorry for this mistake. We have corrected it (Abstract; line 5).

  1. Table 1: The caption "W; weeks" appears on the next page (line 140).  It should be moved to be part of Table 1.  Also, a description of what the bold values mean (statistical significance) should be added to Table 1 and 2 and 3.

Answer:

   Thank you for the important suggestions. In the revised version of the manuscript, we have modified Table 1. In addition, we have added a description of what the bold values mean into Tables 1 – 3.

  1. Line 175 (Begin sentence with "Another study showed..." or "A previous study showed..."), Line 184 (subjects.) and Line 206 (that a honey bee product…).

Answer:

   Based on your suggestion, we have modified the indicated texts in 4. Discussion (1st paragraph, line 4), (2nd. paragraph, line2), and (3rd. paragraph, line11). Furthermore, we have re-checked all sentences for grammar and style.

Reviewer 2 Report

The manuscript by Yuno et al describes the analysis of 33 patients from a previous RCT of patients with renal cell carcinoma. The manuscript represents the third report of the same (small) patient cohort treated with royal jelly, and as such, the difference to their previous reports is incremental. The differences to their previous reports are: additional time points between 0-3months, and analysis of MCSF. Most data points that show significant differences (at 3 months) were reported before (MCO 2020). Nevertheless, this RCT appears novel and additional/repeated analyses of the data may be warranted.

The authors need to explain the differences to the original trial protocol, eg the protocol called for recruitment of 60 patients (n=33 were included in all 3 reports – what happened to the rest?). Also, why was the dose of royal jelly increased from 2400 mg/day to 3600 m/day?

Detailed comments:

No need to abbreviate adverse events or royal jelly.

Abstract as a stand-alone is not clear.

Explanation for pre/post ratios unclear: is the pre-treatment data the same as patient baseline data, ie prior to administration of royal jelly?

Was a correction done for multiple comparisons?

What main type of RCC were these patients? ccRCC?

Data for 3 months in Figure 1 and table 2 were reported in MCO 2020.

Tables require footnotes to detail what numbers mean (IQR, 95% CI or range?)

The description of macrophages, as being M1 or M2, is oversimplified and outdated – it is now accepted that TAMs exist on a spectrum.

Author Response

August 28th, 2020

Reviewer 2

Comments to reviewer 2:

We thank the reviewer for evaluating our manuscript. We agree with your opinions, and your advice have helped us to greatly improve the manuscript. Our responses to your comments are provided below (page and line numbers in the revised version of the manuscript are indicated). In addition, the manuscript has been proofread by a native English-speaking editor from a professional English language editing service.

Point-by-point response

  1. Most data points that show significant differences (at 3 months) were reported before (MCO 2020). Nevertheless, this RCT appears novel and additional/repeated analyses of the data may be warranted.

Answer:

   Thank you very much for this important suggestion. As you pointed out, we previously reported the clinical roles, including the anti-cancer effects and preventive effects on adverse events, of oral intake of royal jelly in renal cell carcinoma (RCC) patients treated with tyrosine kinase inhibitors (TKIs) (K Araki et al. Medicines 2018). However, in that study, the detailed molecular mechanism of such RJ-related functions was not clear. We also previously showed the relationship between such TKI-induced anorexia and fatigue and changes in serum levels of TNF-α, TGF-β, and IL-6 at 3 months after treatment in a similar study population (Y Miyata, et al. Molecular and Clinical Oncology, 2020). In that study, we found that changes in serum levels of TNF-α and TGF-β played important roles in such RJ-related suppression of anorexia and fatigue.

In the present study, we paid special attention to M-CSF because TNF-α, TGF-β, and IL-6 were produced from tumor-associated macrophages in RCC. In addition, we also newly analyzed temporal changes in MSCF, TNF-α, TGF-β, and IL-6 from 2 weeks to 12 weeks because serum levels of these inflammation-related factors are quickly changed by various stimuli including drugs and external environments, and then, such phenomena lead to changes in pathological activities of TNF-α, TGF-β, IL-6, and MCSF.

Thus, although we investigated 3 issues in the same study population and used the same study protocols, the purpose and results of the present study are different from those of our two previous studies. Finally, we have emphasized the novelty of this study and how if differs from 2 previous reports in the revised version of the manuscript (Abstract, last sentence; 1. Introduction, 4th. Paragraph, lines 2-3, 9-10, 14-17).

  1. The authors need to explain the differences to the original trial protocol, eg the protocol called for recruitment of 60 patients (n=33 were included in all 3 reports – what happened to the rest?). Also, why was the dose of royal jelly increased from 2400 mg/day to 3600 m/day?

Answer:

   The protocol and study population in this study were similar to those in our pervious reports. In fact, number of patients was 33 in all 3 studies.

Regarding the dose of royal jelly, we indicated it in the two previous reports as follows: “RJ and the placebo were prepared as capsules containing 900 mg RJ and starch, respectively, that share the same taste, smell, size, shape, and color. Capsules were orally administered four times per day for three months” (K Araki et al. Medicines 2018) and “RJ and placebo were prepared as capsules containing 900 mg RJ and starch, respectively. They were similar in taste, smell, size, shape, and color. Capsules were administered orally four times per day (after breakfast, lunch, and dinner and before bedtime) for three months” (Y Miyata et al. Molecular and Clinical Oncology 2020). Thus, all doses of royal jelly in the 3 studies were 3,600 mg.

Finally, we have modified some sentences about the study protocol and dose of royal jelly for clarity (2.2. Patients and study population; lines 1, 7-8)

  1. No need to abbreviate adverse events or royal jelly.

Answer:

   According to your suggestion, we have deleted the abbreviation for adverse event (AE) from the text (Abstract). On the other hand, we still used abbreviation of royal jelly (RJ) because RJ was often used in the text.

  1. Abstract as a stand-alone is not clear.

Answer:

   Thank you for your important suggestion. We have modified the Abstract to clarify our results and the new findings of this study (lines 15-20).

  1. Explanation for pre/post ratios unclear: is the pre-treatment data the same as patient baseline data, ie prior to administration of royal jelly?

 Answer:

   Yes. The pre-treatment data were collected in the morning of the first treatment day and were evaluated before oral intake of RJ and TKIs. We have added this information to 2.2. patients and study protocol (last sentence).

  1. Was a correction done for multiple comparisons?

Answer:

   In this study, all comparisons of data were performed between royal jelly group and placebo group or between absence of anorexia group and presence group in each treatment period. In short, our study has no data on comparison of these data among 2, 4, 6, 8, 10, and 12 weeks. Therefore, we believe that multiple comparison is not necessary in this study design. On the other hand, we are afraid that a sentence in the Figure legends of Figure 1 (TNF-α between 6 and 12 weeks) may be misleading. Therefore, we have modified this sentence in the Figure legends of Figure 1 (line 4-5).

  1. What main type of RCC were these patients? ccRCC?

Answer:

   Our study population included 29 clear cell (conventional) carcinoma (87.9%) patients. However, there was no significant difference in the rate of clear cell carcinoma. Your question is important to understanding our results and discussion. Therefore, we have added this information to 2.2. patients and study protocol (lines 3-5).

  1. Data for 3 months in Figure 1 and table 2 were reported in MCO 2020.

Answer:

   Your opinion is correct. As shown in this response, one of the main purposes of this study is to clarify the temporal changes in inflammatory-related factors by oral intake of RJ. Therefore, we thought that duplication of a part of data in Figure1 and Table 2 may be permitted. However, to avoid misunderstanding, we have deleted our previous data on TNF-α, TGF-β, and IL-6 at 12 weeks (Y Miyata, Molecular and Clinical Oncology 2020) from Figure 1. We have added sentences about them instead into the Results section (3.3. Changes in the post-/pre-treatment ratio of inflammation-related markers after RJ intake, lines 3-5, 7-9, 11).

On the other hand, we have added comments about citation of data at 12 weeks from our pervious report [7] in Table 2. At first, we deleted the data at 12 weeks from Table 2 and added the related information to the Discussion. However, we are afraid that temporal changes in the relationships between post-/pre-treatment ratio of inflammation-related markers and anorexia or fatigue become unclear if data at 12 weeks are omitted from the Results section. In addition, a similar Table was not included in our previous reports, and we did not show the results after 12 weeks in the text. Therefore, we have modified this. If you and the Editors strongly recommend deleting the data at 12 weeks from Table 2, we will do it immediately.

  1. Tables require footnotes to detail what numbers mean (IQR, 95% CI or range?)

Answer:

   Thank you for your suggestion. Numbers in Table 1 and 2 were means median / interquartile range (IQR). We have added such information to Tables 1 and 2. Regarding Table 3, we have indicated it in the title.

  1. The description of macrophages, as being M1 or M2, is oversimplified and outdated – it is now accepted that TAMs exist on a spectrum.

Answer:

   We agree with your opinion. However, we think that such concept of M1 and M2 is still accepted (Nature Commun 2020; 11: 4035; Allery 2020 in press; J Immnother Cancer 2020; 8: e 000778; Cells 2020; 9: 1535). On the other hand, we also understand that there is objection to this classification of macrophages as M1and M2 (Murray PJ et al. Immunity 2014; 41: 14-20). In addition, there is a general agreement that the detailed mechanisms of TAM turnover, function, and ontogeny are complex and are not fully understood (Laviron M et ak, Front Immunol 2019; 10: 1799). Such information is also important to discussing our results. Therefore, we have added these comments and cited the previous reports in the Discussion (last sentence, reference [38, 39]) as limitations. Thank you very much for your important opinion.

Round 2

Reviewer 2 Report

The authors have largely addressed my concerns. below are a few comments that may help further improve the manuscript.

Instead of using 'conventional RCC' the authors should use 'clear cell RCC'.

I would prefer leaving the 12 week data in the figures but clearly mention in the legend that this has previously been reported.

The two new references have formating and spelling errors.

The authors misunderstood my comment regarding 'explain the differences to the original trial protocol'. I was referring to the clinical trial protocol  (https://upload.umin.ac.jp/cgi-open-bin/ctr_e/ctr_view.cgi?recptno=R000023278), which states: 

'The group of royal jelly patients take 2400mg /day royal jelly for 90 days.'

'

Target sample size 60

A comment with this in mind would be useful.

Author Response

Response to Reviewer 2

Thank you for evaluating our manuscript. We appreciate your positive feedback. We believe that your suggestions have helped us to improve the quality of our manuscript.

Our point-to-point responses to your comments are provided below. Line numbers indicated in parenthesis refer to the revised manuscript. In addition to this point-by-point response, the revised version of the manuscript is attached as a separate file.

Your comment 1: Instead of using 'conventional RCC' the authors should use 'clear cell RCC'.

Response)

We thank you for your suggestion. Based on your suggestion, we used “clear cell RCC” in all text (2.2. Patients and study protocol, lines 3 and 4).

Your comment 2: I would prefer leaving the 12 week data in the figures but clearly mention in the legend that this has previously been reported.

(Response)

   Thank you for your opinion. In previous revision, we deleted the 12-week data of TNF-α, TGF-β, and IL-6 from original version of the manuscript according to your suggestion. On the other hand, relationship between royal jelly intake and serum MCSF levels is new data, but not showed in previous report. Therefore, we showed the 12-week data of MCSF in Figure 1. However, as you pointed, I think that your opinion should be shown in Figure legend (last sentence). Therefore, we added the comment “The 12-week data of TNF-α, TGF-β, and IL-6 are showed in previous report [7]” in this revision.

Your comment 3: The two new references have formating and spelling errors.

(Response)

   We are sorry for simple mistake. I modified 2 references (reference 38 and 39)..

Your comment 4: The authors misunderstood my comment regarding 'explain the differences to the original trial protocol'. I was referring to the clinical trial protocol (https://upload.umin.ac.jp/cgi-open-bin/ctr_e/ctr_view.cgi?recptno=R000023278), which states: 'The group of royal jelly patients take 2400mg /day royal jelly for 90 days.'

A comment with this in mind would be useful.

(Response)

   We understood what you want to say in this time. Thank you very much for important opinion. We confirmed the record about administration of royal jelly. As results, all patients took the royal jelly according to our protocol (2,400 mg / day). In short, we showed wrong dose of royal jelly in previous reports. I understand that it is very important mistake. Therefore, we will contact the Editors of our previous reports immediately. On the other hand, we would like to emphasize that all results and comments are not changed in these previous two reports.

On the other hand, we tried to recruit 60 patients in this clinical study. However, unfortunately, it failed to reach the goal. We added such information into 2.1. Ethics (lines 2 – 5).